# Nutritional Counseling Promotes Adherence to the Mediterranean Diet and Healthy Eating in Italian Patients Affected by Phenylketonuria and Treated with Pegvaliase

**DOI:** 10.3390/nu16193348

**Published:** 2024-10-02

**Authors:** Michele Stecchi, Alice Rossi, Michelle Santoni, Francesca Bandini, Lucia Brodosi

**Affiliations:** 1Department of Medical and Surgical Sciences, University of Bologna, Via Zamboni, 33, 40126 Bologna, Italy; michele.stecchi@studio.unibo.it (M.S.); santonimichelle@gmail.com (M.S.); francesca.bandini10@studio.unibo.it (F.B.); 2Clinical Nutrition and Metabolism Unit, IRCCS AOUBO, Via Albertoni, 15, 40138 Bologna, Italy; alice.rossi@aosp.bo.it

**Keywords:** phenylketonuria, Pegvaliase, Mediterranean diet, free diet, nutritional counseling

## Abstract

Background/Objectives: Pegvaliase, a subcutaneous therapy to treat phenylketonuria (PKU), has allowed these patients to maintain adequate phenylalanine (Phe) blood values without following a Phe-controlled diet; this brings up the challenge of promoting healthy eating while moving away from prescription diets. In our center, every patient treated with Pegvaliase undergoes routine nutritional counseling aimed at promoting adherence to the Mediterranean diet (MedDiet) during regular inpatient visits. This study aims to assess adherence to MedDiet and the adequacy of the diets of patients treated with Pegvaliase regarding micro- and macronutrients. Methods: Seven patients on chronic therapy with Pegvaliase underwent a dietetic evaluation to assess the composition of their diets in terms of micro- and macronutrients; they were also administered the Mediterranean Diet Score (MDS) questionnaire. Subcategories from MDS were extracted to evaluate the consumption of foods typically included (vegetables, olive oil, etc.) and typically excluded (red meat, etc.) in the MedDiet. To assess the adequacy of the diet, nutrient and energy levels were compared with guidelines for the Italian population. Results: MedDiet adherence in our sample was comparable to the general population; in terms of macronutrients, good adherence to the recommendations was observed, with every one of them met except for excessive simple sugar consumption. Micronutrient dietary intake was inadequate for zinc, iron, selenium, folate, thiamine, and riboflavin. Conclusions: While more work is necessary to help patients treated with Pegvaliase to progress toward healthy eating, our study suggests that nutritional counseling routinely performed during inpatient visits, typically twice a year, effectively promotes healthier eating habits than those observed in the general population.

## 1. Introduction

Phenylketonuria (PKU) is a rare genetic disorder caused by a deficiency in the activity of the enzyme phenylalanine hydroxylase (PAH), the enzyme responsible for converting phenylalanine (Phe) into tyrosine (Tyr) [1]. The global prevalence of the disorder is close to half a million, with various genotypes and consequent clinical phenotypes; the incidence varies in different countries, ranging from 1:4500 live births in Italy to 1:125,000 in Japan; PKU is the most common autosomal-recessive disorder of amino acid metabolism [2,3]. Left untreated, it can have irreversible impacts on people who are affected by it; the main complications it can lead to are intellectual disability, epilepsy and behavioral disorders [1,3]. Newborn screening (NBS), performed using the dried blood spot during the first days of life [4], allows prompt diagnosis of PKU and, therefore, an immediate start of treatment before high blood Phe might have repercussions on long-term intellectual outcomes.

The cardinal treatment of PKU is dietary, which requires patients to adhere to a diet with reduced PHE content. As Phe is an amino acid naturally present in food, patients with PKU must follow a diet with a low content of natural protein and complementary consumption of protein substitutes to achieve an adequate protein intake. A strict collaboration between the various specialists caring for the patients and metabolic dietitians is necessary to personalize the diet in order to meet individual needs, prevent other comorbidities, and minimize social difficulties [1,5]. In 2007, sapropterin dihydrochloride was approved by the Food and Drug Administration to treat patients with PKU and subsequently approved by the European Medicines Agency in Europe. It is a cofactor for PAH, and its effectiveness varies based on the genetic and clinical phenotypes of patients; in patients with PAH deficiency responsive to sapropterin supplementation, it increases tolerance to PHE content in the diet, allowing for metabolic control in patients with less strict diet management, and it should be noted that treatment with this cofactor is approved for patients of any age [6]. Since 2021, Pegvaliase has been available in Italy for treating PKU in patients at least 16 years old with blood Phe levels above 600 μmol/L (10 mg/L) despite dietary prescription [7]. Pegvaliase is an injective enzyme-substitution therapy, converting Phe to ammonia and trans-cinnamic acid [8]; adequately dosed, it is effective in reducing Phe values in almost every patient, independently of the genetic mutation they might be affected by and their Phe dietary content; this allows maintenance of Phe values at target without following a diet with controlled Phe content [9]. Unequivocally, diet management in patients on Pegvaliase therapy brings up new challenges, the first being protein intake management during dose escalation and the second one, while at a stable dose, controlling Phe levels while avoiding protein malnutrition and hypophenylalaninemia. Worldwide, various experts came together to suggest the correct dietary management for patients taking Pegvaliase, which has resulted in most patients obtaining adequate Phe blood levels with a free diet and no sign of malnutrition [10,11]. The development of new treatments and improvement of traditional ones, associated with efficient NBS, are resulting in patients growing older with better quality of life and less burden of disease, which however, brings up new challenges not necessarily related to a higher than recommended blood Phe value [12,13]. 

In particular, patients affected by PKU appear to be at a higher risk of having lower bone mass density [14] and are more prone to diseases associated with poor metabolic health, such as hypertension, dyslipidemia, and obesity [15,16]. Considering modifiable risk factors for these diseases, lifestyle plays a critical role in prevention while also being crucial in their treatment in the general population [17,18]; in particular, the Mediterranean diet (MedDiet) remains a dietary model that promotes healthy aging, reduces the incidence of the most common comorbidities in Italy (i.e., hypertension, dyslipidemia, cardiovascular disease, type 2 diabetes, cancer) and reduces the risk of overall mortality [19,20]. These findings, first delved into by the pioneering works of Keys et al. [21], must be taken into consideration together with modern views that diets must also be environmentally and economically sustainable while remaining consistent with the local traditions of Italy [22]. 

Managing adequate micronutrient intake has always been challenging while following a diet with controlled Phe content, with biochemical deficiencies commonly reported even when observing adequate micronutrient intake in the diet of patients affected by PKU [23,24,25]. Concerning Pegvaliase, to our knowledge, only one study has evaluated micronutrient intake in patients taking Pegvaliase, but the complete assessment was only performed when patients were starting Pegvaliase, and all but one were still regularly consuming at least one medical product; the authors reported adequate intake for every considered micronutrient but Vitamin D [26]. In Italy, a panel of experts has already published that it is necessary to promote nutritional counseling in guiding the transition from a medically Phe-controlled diet to a regular one while introducing Pegvaliase [27]. In our center, after the induction and titration phase is concluded and Pegvaliase therapy is at the therapeutic dose, inpatient visits are typically scheduled every six months unless specific needs arise. It should be noted that during the induction and titration phases, patients are evaluated more frequently, both by the physician and by the dietitian. In the same setting, the dietician evaluates the eating habits of the patient, and nutritional counseling takes place; this counseling is aimed at promoting a healthy diet based on the MedDiet.

The main aim of this study is to evaluate how patients taking Pegvaliase who have completed both the induction and the titration phases, and who regularly undergo nutritional counseling, structure their diet, particularly concerning healthy eating and their adherence to the MedDiet. Secondly, we aim to evaluate micronutrient dietary intake in relation to age and sex-specific requirements, evaluating the potential need for more specific and structured nutritional interventions.

## 2. Materials and Methods

Patients currently treated at the Clinical Nutrition and Metabolism Unit, IRCCS AOUBO, Bologna, Italy, who are affected by PKU and receiving Pegvaliase therapy, who have completed the induction and titration phases of the drug, and who are on an unrestricted diet were included. In the pre-screening, patients whose food choices are made by the caregiver were excluded.

Data on anthropometric measurements (height, weight, waist circumference) were collected in the morning on an empty stomach, after urination, wearing underwear, and the BMI was calculated by dividing the weight, in kilograms, by the square of the height, in meters. Data from laboratory tests were collected regarding the main metabolic parameters (glucose, insulin, total cholesterol, HDL, triglycerides, uric acid, TSH, PTH, homocysteine), vitamins (vitamin B12, folate, vitamin D), and micronutrients (iron, ferritin, transferrin, calcium, phosphorus, zinc). The homeostatic model assessment (HOMA index) was calculated with the following formula: fasting blood glucose (mg/dL) × fasting insulin (mU/L)/405.

The patients were subjected to a telephone, 7-day dietary recall conducted by a dietitian with expertise in PKU dietary treatment. During the clinical evaluation and according to clinical practice, the dietitian asks the patient to recall what they have eaten in the previous 7 days. This method of assessing diets has clear limitations (as do other methods): the week might not be a standard one, and the patient might not be able to recall precisely what foods or how much of them was eaten. The dietary recall data were uploaded to the program commonly used in our clinical practice, MetaDieta vers. 3.7.1.1 (METEDA S.r.l. ©, San Benedetto del Tronto AP, Italy).

The analysis of the diet allowed us to calculate the following parameters for dietary content: intake of total energy in kcal; and subdivision into macronutrients (proteins, carbohydrates, lipids, and alcohol) and micronutrients (iron, calcium, phosphorus, magnesium, selenium, zinc, vitamin B12, folate riboflavin, vitamin A, niacin, thiamine). Lab parameters were collected from the analyses performed during regular clinical practice (iron, ferritin, transferrin, calcium, phosphorus, magnesium, zinc, glucose, insulin, total cholesterol, HDL, triglycerides, uric acid, TSH, vitamin D, PTH, homocysteine, vitamin B12, folate).

Subsequently, the average daily nutritional intake of the patients was compared to the Italian guidelines, the “IV revision of Nutrients and Energy for the Italian population” (LARN) [28]. For comparison, the Population Recommended Intake (PRI) and the Reference Interval (RI) were used for minerals and vitamins, and for macronutrients, respectively. For comparing cholesterol intake, the Suggested Dietary Target (SDT) was used. Targets were chosen according to age and sex, and patients were grouped based on having the same SDT (e.g., females of fertile age were grouped separately for iron as their SDTs differ).

Patients were also administered the MDS to assess adherence to the MedDiet; the questionnaire has been described previously by Panagiotakos et al. [29] and includes questions on non-refined cereals, fruits, vegetables, legumes, potatoes, fish, red meat, meat products, poultry, full-fat dairy products, olive oil, and alcohol. The scoring system is designed to reflect the typical dietary patterns observed in the Mediterranean region. The total score is assigned a value between 0 and 55, with higher scores reflecting greater adherence to the MedDiet. To calculate the score, the weekly consumption of foods is considered, from 0 to at least 5 times per week. For foods that are part of the MedDiet (non-refined cereals, fruits, vegetables, legumes, olive oil, fish, and potatoes), more frequent consumption correspond more points, up to 5 times per week for maximum points; on the other hand, for foods that are excluded from the MedDiet (red meat and products, poultry, and full-fat dairy products), higher scores correspond to lower frequency of consumption, with the highest score corresponding to not consuming those foods on a weekly basis.

We also decided to evaluate the tendency to consume particular food groups; in particular, extrapolating data from the MDS, we established various subgroups, similar to the approach of Bacharaki et al. for patients with end-stage kidney disease [30]. We established the subgroup “Avoid foods”, comprising foods not traditionally found in the MedDiet (poultry, red meat and products, full-fat dairy products), and the group “Recommended foods”, which includes the pillars of the MedDiet (non-refined cereals, potatoes, fruits, vegetables, legumes, fish, olive oil); in addition to this, we considered the subgroup “Fruit, vegetable and legumes” (FVL). Each subgroup score was the sum of the score of the single items considered. For “Recommended foods”, the score was a value ranging from 0 to 35; for the subgroup “Avoid foods”, the range was from 0 to 15, and for the subgroup “FVL”, from 0 to 15. Values were expressed as percentages. A similar application of the MDS. Continuous variables were expressed as median, minimum, and maximum. Categorical variables were expressed as absolute frequency and percentages. The associations were evaluated with Spearman’s rank correlation coefficient. Statistical significance was set at *p* < 0.05.

Statistical analysis was performed using StatView (Version 5.0.1) for Windows 92-98, SAS Institute (Lane Cove, Australia). 

The data analyzed belonged to patients enrolled in the “RarEMIX-2021” observational study. This study was approved by the local Ethics Committee (reference code 796/2021/Oss/AOUBo).

## 3. Results

Of nine patients currently being treated with Pegvaliase at our center, two did not meet the criteria, as one was in the titration phase and one was unable to make independent food choices. The final sample of seven patients comprised five men and two women on Pegvaliase therapy for a median of around 30 months (891 days), ranging from 5 to 38 months (150 to 1152 days).

The description of the sample is shown in Table 1.

These were adult patients aged between 18 and 42, with a median BMI indicative of normal weight. The glycometabolic parameters were in the range of what is considered clinically normal; in particular, no insulin resistance was present in our patients. Blood iron and vitamin levels, thyroid function, and micronutrients were, considering the median values, in the range of normality, and there were no clinical signs of pathology.

The nutritional evaluation (Table 2) shows the patients had a median caloric intake of 51% calories from carbohydrates, 16% from proteins, and 34% from lipids. Specifically, the percentage of calories provided by simple sugars was 13% of the total caloric intake, and that of saturated fats was 9%. The consumption of simple sugars and lipids exceeded the recommended guidelines. One patient had a high alcohol consumption. On average, protein consumption was 1 g/kg, which aligns with the Italian recommendations, with a significant portion being of high biological value, as expressed by the ratio of protein coming from animal products or vegetable, ranging from 43% to 69.6%. Regarding micronutrients, we report unsatisfactory zinc, selenium, thiamine, riboflavin, and folate intake. Iron intake was found to be inadequate in the female population, considered their greater daily needs.

Data concerning patients’ adherence to the MedDiet are presented in Table 3. 

Using the Spearman correlation test, we examined whether dietary parameters could be associated with laboratory parameters. Unfortunately, we did not find statistically and clinically significant correlations. The only significant correlation that emerged from the analysis was the positive one between age and BMI (*p* = 0.0358).

## 4. Discussion

In this study, we assessed the composition of the diet of patients affected by PKU who were being treated with Pegvaliase, who had concluded the induction phase, and who were no longer prescribed low-Phe diets; we investigated their adherence to the MedDiet and how adequate their diet was in terms of nutrient intake.

Concerning adherence to the MedDiet, the median adherence was 49.1%, with the highest adherence observed being 61.8% and the lowest being 34.5%. It is important to underline how MDS considers alcohol consumption; in particular, it attributes 0 points to people who do not consume any alcohol, as the questionnaire considers that people who consume moderate amounts of alcohol have the greatest adherence to MedDiet. Considering the original publication from Panagiotakos et al., the median adherence we observed was 27, in line with the average reported by them, which was 26 [29]. Considering the subcategory “recommended foods”, whose regular consumption is part of the MedDiet, the median percentage value observed was 62.9% (ranging from 42.9 to 74.3); median scores of 5 were observed for olive oil, unrefined cereals, and vegetable consumption, indicating high adherence; the median scores for fruit and fish were 3 and 2, respectively, indicating moderate regular consumption; the median scores for potatoes and legumes were both 0. Considering the value of 62.9%, consuming foods that are part of the MedDiet was a significant positive contributor to the median percentage score of MDS, which was 49.1%. Every patient considered in the study was familiar with the low-Phe diet, as they had been previously prescribed this diet or tailored their diet accordingly, at least in childhood; potatoes and legumes are pivotal sources of natural protein that every patient had structured their diet around. This resulted in frequent consumption while following a Phe-controlled diet [5]. This factor might play a role in lower consumption of these food groups by the patients after the liberalization of their diets; the same assumption can be made when evaluating the subgroup FVL, whose median value was 60%, which is in line with the category above. 

On the other hand, some foods almost always excluded in Phe-controlled diets were part of the “avoid foods” subcategory. The considered sample displayed high consumption of these foods, as can be seen by the median value of 26.7%. It is not easy to speculate on the psychological impact of removing a previously established medical prohibition to consume certain foods. A previous systematic review considered patients with PKU who started sapropterin dihydrochloride therapy, allowing them to free their diet. The authors evaluated the need for nutritional education to avoid the risk of inadequate diets, but adherence to the MedDiet was not evaluated [31,32]. In general, adult patients with PKU show low adherence to a Phe-reduced diet into adulthood and to clinical follow-up in general [33]. It is logical to speculate that the nutritional counseling to adhere to a healthy diet is a weaker medical veto compared to the previous requirement to follow a diet with a controlled phe content. Comparing our results with those previously reported by Cazzorla et al. in a population of adult patients with PKU not treated with Pegvaliase, the consumption of foods rich in protein not included in the MedDiet is typically high in general in this population, even without medical indications to lift the dietary restrictions [34].

Overall, concerning MedDiet adherence in our population, we observed that nutritional counseling according to clinical practice, typically performed twice per year during scheduled inpatient visits, effectively promoted adherence comparable to the general population in the north of Italy [35]. In general, our population presented various risk factors for relatively lower adherence to MedDiet, in particular, being young and being from the north of Italy [36]; in addition, as was previously demonstrated, psychological factors such as family stress, which the diagnosis of PKU itself creates [37], correlate to lower adherence to the MedDiet [36]. The MedDiet, recognized by the United Nations Educational, Scientific and Cultural Organization (UNESCO) as an Intangible Cultural Heritage of Humanity, has conviviality as an integral part of the diet itself [38]. While our questionnaire did not evaluate this aspect of sharing meals, without Pegvaliase therapy, it would not be feasible to imagine patients with PKU freely enjoying meals with people who do not have the disease and adequately manage their blood Phe levels. As nutritional counseling has proven to be effective in improving adherence to the MedDiet in the general population [39], more structured programs are necessary for patients on Pegvaliase who switch to a free diet; in particular, such programs should promote the regular consumption of “recommended foods” while reducing consumption of the “avoid foods”. Concerning how the diets of the considered sample were structured, we observed an adequate median calorie percentage coming from lipids, as well as adequacy in the median calories coming from saturated fats; the observed percentage of total lipids was close to the upper limit secondary to high consumption of extra virgin olive oil. Conversely, we noted a significant variability in the consumption of both fats and sugars. Some patients adhered to the recommended daily targets, while others significantly exceeded them. This diversity in dietary habits within the sample is an intriguing aspect of our findings. While there is some elasticity in terms of diet composition [40], the evidence that simple sugars should account for less than 10% of the daily energy intake still holds up robustly, suggesting the adequacy of this recommendation [41]. Concerning saturated fats dietary intake, even though an ever-growing body of literature is reevaluating the impact on overall health [42], as a Unit of Clinical Nutrition, we follow the various Italian guidelines that underline the need to have controlled consumption [28]. Considering protein intake, the median intake of protein per kilogram was 1.0, ranging from 0.8 to 1.1. These data confirm that all the patients were able to adhere to adequate protein intake from natural sources, without either exceeding or falling below the individual needs.

Comparing our findings to those reported for the diet of the general population of the north of Italy [18], considering the median values of the diet composition, our sample meets the recommended nutritional targets for all considered items more closely than the general population in the north of Italy. Considering micronutrient intake, we observed a dietary median intake lower than that recommended for zinc, selenium, folate, riboflavin, and thiamine. Comparing our data with observations by Filippini et al. on the average daily intake for people living in the north of Italy [43], we observed comparable intake concerning zinc. At the same time, our population displayed a significantly lower selenium intake. Selenium dietary deficiency is common worldwide and has been previously observed in Italy, even in a particular population [44]. More studies are necessary to delve into the necessity of chronic supplementation. In general, even though nutritional counseling promotes adherence to the MedDiet, which should adequately meet the daily needs for macro- and micronutrients [45], more tailored and specific nutritional interventions should be performed based on specific individual needs to promote consumption of the specific food groups that might be lacking. For example, considering our population, low to zero consumption of certain foods, such as nuts or shellfish, was observed; promoting consumption of these foods could help meet most of the dietary recommendations still left unmet while adhering more closely to the MedDiet. Considering the clinical and laboratory characteristics of the sample, despite the theoretical micronutrient and vitamin dietary deficiencies, the values we last observed during clinical practice were in the range of normality for all blood parameters, which could be accounted for by the supplementation they had potentially received previously. 

This study has several limitations; the first is the small number of patients we considered, which is to be expected considering PKU is a rare disease and Pegvaliase is not the only treatment. Secondly, the population was heterogeneous in terms of age, background, comorbidities, and duration of treatment with Pegvaliase. Thirdly, the diet considered in this study was evaluated only once and during summer; this might not allow us to capture the actual average diet during the whole year.

## 5. Conclusions

This is the first study that evaluates the adherence to the MedDiet in patients with PKU being chronically treated with Pegvaliase and who undergo regular nutritional counseling; this is also the first study that evaluates the diet composition and the meeting of dietary intakes.

While our study suggests that nutritional counseling effectively promotes adherence to the pivotal principles of the MedDiet and leads to nutrition as healthy as diets observed in the general population, more work is needed to support patients in their journey toward healthy eating, addressing individual needs.

## Figures and Tables

**Table 1 nutrients-16-03348-t001:** Clinical and laboratory characteristics of the sample.

	Median (Min–Max)	Reference Values
Age (years)	33 (18–42)	
Height (cm)	176 (150–185)	
Weight (kg)	73.5 (64.8–85)	
BMI	24.53 (21.04–29.02)	
Biochemical parameters		
Glucose (mg/dL)	71 (62–80)	60–110
Insulin (mU/L)	7.3 (3–11.3)	1.9–23.0
HOMA (mU/mmol)	1.3 (0.48–1.98)	<2.5
Total cholesterol (mg/dL)	154 (111–218)	<200
HDL cholesterol (mg/dL)	51 (46–57)	>35
LDL cholesterol (mg/dL)	74 (41.2–146.2)	<116
Triglycerides (mg/dL)	50 (37–279)	<150
Uric Acid (mg/dL)	4.3 (3.9–5.1)	3.4–7.0
PTH (pg/mL)	27 (21–51)	12–88
Homocysteine (micromol/L)	11.1 (9.2–39.9)	5.0–15.0
TSH (mUI/mL)	1.2 (0.6–2.6)	0.25–4.50
Iron (ug/dL)	90 (49–121)	70–180
Transferrin (mg/dL)	242 (208–265)	200–360
Transferrin saturation %	28.7 (14.7–41)	15–55
Ferritin (ng/mL)	43 (28–240)	24–336 (M)
Calcium (mg/dL)	9.4 (8–10.5)	8.6–10.5
Phosphorus (mg/dL)	3.1 (2.4–3.5)	2.5–4.5
Zinc (ug/dL)	95 (36–110)	70–120
25-hydroxy vitamin D (ng/mL)	26 (9–45)	<12
Vitamin B12 (pg/mL)	211 (168–493)	145–914
Folate (ng/mL)	3.7 (2–5.4)	3.1–19.9
Phe (micromol/L)	267.92 (109.45–674.5)	<600
Tyr (micromol/L)	27.17 (21.58–68.92)	<250

HOMA: Homeostasis model assessment; HDL: high-density lipoprotein; LDL: low-density lipoprotein; PTH: parathyroid hormone.

**Table 2 nutrients-16-03348-t002:** Diet composition obtained through nutritional recall.

	Median (Min–Max)	Suggested by Guidelines
Energy (Kcal/day)	1970 (1474–2550)	
Carbohydrates (%/Kcal)	51.4 (26.1–53.6)	45–60%
Simple sugars (%/Kcal)	13 (6.6–23.3)	<10%
Proteins (%/Kcal)	15.9 (9.2–19.1)	12–15%
Lipids (%/Kcal)	34.1 (26.8–37.7)	25–30%
Saturated fat (%/Kcal)	8.8 (7.6–10.8)	<10%
Alcohol (g/day)	0.06 (0–140)	NA
Fiber (g/day)	16 (11–21)	15–25 g
Cholesterol (g/day)	202 (141–335)	<300
Protein pro kg	1 (0.8–1.1)	0.8–1 mg/Kg
Animal proteins (g/day)	39 (37–54)	NA
Vegetal proteins (g/day)	28 (16–49)	NA
Animal protein/vegetal protein (%)	61.7 (43.3–69.6)	NA
Calcium (mg/day)	1342 (918–155)	134% (92–155%)
Phosphorus (mg/day)	1095 (836–1311)	156% (119–187%)
Iron (mg/day)	10 (7–20)	83% (49–201%) *
Magnesium (mg/day)	277 (191–394)	115% (80–164%)
Zinc (mg/day)	9 (8–10)	84% (74–102%) *
Selenium (mg/day)	37 (28–56)	68% (50–102)
Thiamine (mcg/day)	0.8 (0.7–0.9)	70% (64–78) *
Niacin (mg/day)	20 (14–22)	109% (80–121)
Riboflavin (mg/day)	1.3 (0.8–1.6)	84% (52–118) *
Vitamin B12 (mcg/day)	3.5 (2.7–7)	144% (114–290)
Folate (mcg/day)	335 (196–385)	84% (49–96)
Vitamin A (mcg/day)	854 (383–1140)	126% (55–190) *

* Percentage calculated on the recommended value for sex.

**Table 3 nutrients-16-03348-t003:** Responses to the MDS questionnaire.

	Median (Min–Max)
Unrefined cereals	5 (1–5)
Potatoes	1 (0–4)
Fruit	3 (0–5)
Vegetables	5 (3–5)
Legumes	0 (0–1)
Fish	2 (1–3)
Red meat and products	2 (0–4)
Poultry	2 (0–3)
Full-fat milk and dairy products	0 (0–3)
Olive oil	5 (5–5)
Alcohol	0 (0–5)
Tot	27 (19–34)
Adherence (%)	49.1 (34.5–61.8)
Avoid foods	4 (2–8)
Avoid foods (%)	26.7 (13.3–53.3)
Recommended foods	22 (15–26)
Recommended foods (%)	62.9 (42.9–74.3)
FVL	9 (3–11)
FVL (%)	60 (20–73.3)

Scoring for each item ranges from 0–5, 5 being representative of higher adherence to the MedDiet. FVL: Fruit–Vegetables–Legumes. Recommended foods: non-refined cereals, potatoes, fruits, vegetables, legumes, fish, olive oil. Avoid foods: poultry, red meat and products, full-fat dairy products.

## Data Availability

Data are available upon request to the corresponding author. The data are not publicly available due to privacy reasons.

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
