# Peer review of "Nutritional Counseling Promotes Adherence to the Mediterranean Diet and Healthy Eating in Italian Patients Affected by Phenylketonuria and Treated with Pegvaliase"

_nutrients, 2024, doi:10.3390/nu16193348_

Round 1

Reviewer 1 Report

Comments and Suggestions for Authors

Dear Authors

Considering English is not your first language this is a good attempt but unfortunately the English does need a lot of work and editing. I have provided a break down of the difficulties I found when reading this paper.

It is an important piece of work but does need to be edited and the limitations addressed. 

Comments on the Quality of English Language

The English needs to be more concise there are many sentences that could be condensed and made shorter. There are sentences that do not  make complete sense, and some are open end sentences. 

Author Response

Dear Editor,

we would like to acknowledge the reviewers for providing useful and constructive comments to our manuscript. We carefully revised the text according to the numerous reviewers’ suggestions and comments and we believe that the article improved a lot since its initial version.

Reviewer 1.

Review: Nutritionally counselling is effective in promoting the Mediterranean diet and healthy eating in Italian patients affected by PKU treated with Pegvaliase

Overall

This is an interesting paper, however, one of the main criticisms is the English language and its structure- there are sections that are repetative. The paper could be reduced in size and the language for some sections more concise.

We revised English and tried to be more concise.

Additionally there are a number of questions that need to be addressed

Comments

Abstract:

  1. Line 21 Subcategories from MDS were extracted could be re worded subcategories from the MDS were divided into foods that were recommended (e.g,,) and those avoided e.g. The reader is not aware of what foods are recommended / avoided

We made clearer what is and what isn’t part of the MedDiet according to MDS questionnaire.

  1. Line 28/29: Nutritional counselling (spelling) and you need to state in your abstract how intense this was. If counselling was weekly etc

We modified the article accordingly. The nutritional counseling follows a variable schedule, as in happens in conjunction with the medical inpatient visit. In particular, as the patient introduces Pegvaliase, patients typically are evaluated more frequently and therefore the nutritional counseling is more frequent. Progressively, inpatient visits become more far apart, and during chronic therapy we evaluate patients tipically twice a year (therefore patients undergo the nutritional counseling with the same frequency).

The word “counselling”, can be spelled “counseling” as well, as reported in Oxford dictionary (https://dictionary.cambridge.org/dictionary/english/counseling). We changed the title accordingly.

Introduction

  1. Line 40/41 needs to be reworded as does not make complete sense e.g. It can irreversible impact people who are affected by it, among other complications it can potentially not adequately treated… This does not flow, what are ‘other complications’

Thank you for pointing this out, we missed this during the proofreading of the draft.

  1. Line 43/44 the English needs to be reworded and reference is made to avoid consequences but these are not stated. This sentence could be re worded to state something along the lines. Newborn screening allows access to immediate treatment started within the first 10 days of life, avoiding the consequences of long term intellectual outcome etc

Thank you for pointing this out, we missed this during the proofreading of the draft.

  1. Line 46/47 you mention a PHE content but you need to state what is a PHE diet and within this mention protein substitutes.

We added the line you suggested on protein substitutes.

  1. Line 49 and 50 FDA and EMA need to be in full please

Done

  1. Line 60 to 62 These few sentences could be more succinct , also protein substitutes should be mentioned earlier, and they do more than supplement nutritional deficits

As you suggested this part was made more concise.

  1. Line 73 there are 2 systematic reviews which state bone mass is not at risk in PKU patients it states bone mass is lower than the general population but is within clinically normal parameters

We agree with the Reviewer, therefore our aim was just to point out that there is a risk for a lower bone mass in patients with PKU.

  1. Line 79 comorbidies state what these are

Text was edited accordingly, stating the most common chronic conditions in the Italian population that are positively impacted by adhering to the MedDiet .

  1. Line 83 and this paragraph not sure if need this as it is repeated previously. I would try to summarise the unique challenges that Pegvalise brings to older PKU patients with an established dietary pattern. As a group rarely eating high biological protein. What would be valuable is a table in the results section highlighting typical diet pre and post treatment

We changed it, so that it was less repetitive.

  1. Line 98 6 months this is not regular contact for an intensive dietary change

Text was edited. No patient underwent intensive nutritional counseling for this study. This study considers the effectiveness of the standard of practice in our center is concerning adherence to a healthy diet in patients with PKU.

MATERIALS AND METHODS

  1. Line 111 not sure if suffering is the correct word

We corrected it.

  1. Line 118 not sure if need to state how to calculate BMI as this is a standard measurement

We decided not to make any change concerning BMI because we received such observation by Reviewer for previous articles.

  1. Line 122 HOMA (expand definition)

Corrected

  1. Line 123 7 day recall this needs to be explained in more detail please. To review a diet after 7 days would be very inaccurate. In your limitations you do need to state that any dietary method is inaccurate

Text was edited to make it more clear.

  1. Line 137 need to define groups to say as necessary is too vague

Text was edited to be more clear.

  1. Line 143 This section could be shorter and include a shorter explanation of MDS … A high MDS (> 30) is favourable and includes unrefined cereals, fruits, vegetables, legumes etc A low MDS (e.g. <30) is detrimental and includes refined cereals, sugar etc. I assume that some lean red meats are encouraged as a method of iron intake?

We rewrote the part on the scoring, explaining how it works. The MDS, as a questionnaire which we decided to use, only considers the items presented in text before the part on scoring. Simple sugars are not considered by it, nor are refined cereals. Similarly, for red meat, the MDS considers it as not being part of the MedDiet; consumption of red meat less than once per week corresponds to the greatest score, with the score lowering as the consumption increases. 

  1.  A shorter explanation of food groups is also needed, unfortunately the English in this section needs revising e.g. we established the group ‘avoid foods’, constituted of foods that are not considered part of the Med Diet- I thought poultry was part of the Med diet? Some of this section is also repetitive

We tried to make it clearer. Concerning poultry, rate of consumption is typically variable. The MDS, the questionnaire we used, considers it something not to be consumed on a weekly basis to score the maximum amount of points. Scoring for full-fat milk and dairy products and red meat by the MDS works the same way.

Q1. Ideally it would have been good to compared diets pre and post to see what changes were made, and to assess the new foods introduced. This is not discussed and would be valuable to see the adjustments made

This study has a cross-sectional design. We agree that data on before and after the introduction of the drug is interesting, and we plan to do more work to delve into these aspects.

       Q2.  There is no comment from the patients on their new dietary freedom

This study has a cross-sectional design. We agree that data on quality of life is interesting, and we will try to collect data on this as well moving forward.

      Q3.There is no pre and post Phe Tyr levels which would be good to see

This study has a cross-sectional design. We agree that data on quality of life is interesting, and we will try to collect data on this as well moving forward.

Results

  1. Line 171 this could be shorter ie 2 patients failed the criteria one was in the titration phases and one unable to make independent food choices.

Corrected as of your suggestion.

  1. Line 174 nos of months would be good please

Added months, as of your suggestion.

  1. Line 179 ? typo minera

Martial= of iron. We corrected it

  1. Table 1 needs reference values so can see what parameters are within target

Reference values were added.                 

  1. Line 190 protein consumption would be good to give high biological value of protein where is this coming from. Although meeting WHO recommendations for protein what are the sources. If these are all low biological value this shows little change in dietary habits

We tried to address your concern using the ratio of protein coming from animal and vegetable sources. 

Q4. In your abstract it is stated that the patients are adherent to a Med diet, yet it is not mentioned that there are significant dietary inadequacies zinc, selenium, thiamine etc

Abstract has a limited number of words so we had to make a difficult decision on what to include.

Q5. Is it possible from your results to do a table of the foods you state are recommended and those not recommended and give some indication of the frequency of these foods and how much is eaten. This would give some quantification on dietary intake – something along the suggestion

Concerning this, we think that the data on the dietary structure as expressed in Table 2 “Diet and composition obtained through nutritional recall” and Table 3 “Responses to the MDS questionnaire” is enough information to assess the composition of the diet and how it is structured, especially considering we think they support  the study itself.

 Table 2 is the animal / vegetable protein a percentag

Edited accordingly.

  1. Table 3 avoid and recommended foods these need to be clearly defined for the reader. Need ADH% in full please or in the table at the bottom for a definition

Edited.

Q6. Anthropometry would be useful pre and post changes

This study has a cross-sectional design. We agree that data on variation of anthropometry is interesting, and we will try to collect data on this as well moving forward.

Discussion

  1. Line 212 regarding alcohol not sure of the purpose of this sentence – what about the 50% who did take alcohol? How was this accounted for?

We clarified how the MDS questionnarie scores for alcohol consumption.

       Q7. Need to comment that potatoes and legumes were 0 when they would have eaten potatoes previously? I understand that this might be avoidance but what was it replaced by? There is no mention of pasta which is a staple of the Italian diet?

Pasta comes from non-refined cereal, which are considered accordingly to the MDS. This is stated in the material and methods, and we made it easier to be understood.

  1. Line 232 and onwards- this section needs tobe more structured. I understand what you are conveying but it needs to be more concise

Restructured it, making it more concise.

  1. Line 232 and onwards- this section needs tobe more structured. I understand what you are conveying but it needs to be more concise

Rewrote this part according to your comment

Q8 Line 249 regular is not 6 monthly this is only twice a year and not regular

Edited accordingly.

  1. 51 Is this contraindicating your conclusion here to state our population presented various risk factors for a low adherence to MedDiet. I am not sure why Northern Italy would be a risk factor. If you were on a strict PKU diet it would be impossible to comply to a Med diet

Being from the North of Italy is a risk factor according to Ruggiero et al. You are correct that we should’ve cited the article there, and not only at the end of the sentence, and so we did. We also edited the part about PKU and adherence to MedDiet.

       Q9. Line 263 oil is mentioned but has this changed given this is not restricted in a PKU diet?

This study has a cross-sectional design. We agree that data on variation of diet composition before and after Pegvaliase is interesting, and we will try to collect data on this as well moving forward.

  1. You need to review the nutritional intake with plasma values

This is the original comment on plasma values of micronutrients routinely evaluated: the values we last observed during clinical practice were in the range of normality for all blood parameters, which can be accounted for by the potential supplementation they had previously received.

        Q10. Any indication on how patients found it to adjust to dietary changes e.g. tasting meat/ fish

This study has a cross-sectional design. We agree that collecting these data would’ve been interesting, and we will try to do so moving forward.

       Q11. Phe and Tyr levels pre and post diet

This study has a cross-sectional design. We added Phe and Tyr at the time of dietary recall.

     Q12. Discussion on red meat as this is a valuable souce of iron and if not consumed where is this sourced from, similarly diary foods and calcium. It might be good to give a brief overview of nutritional risk factors in changing from a PKU to free liberalised diet

We decided to consider the MedDiet, which is a diet in which red meat is scarcely consumed; in particular we decided, for standardization, to use a validated questionnaire, the MDS.

Limitations

  1. In addition to small nos

Already addressed in the original text

  1. No biochemical data to match dietary intake for all measured nutrients B12, selenium etc.

Almost all of the micronutrients we considered in the nutritional assessment were also assessed on blood, and presented in Table 1.

  1. No interviews with patients to see how they have reacted to the changes- a psychological or QoL questionnaire would have been useful

We agree on this, and we are trying to include in the team someone who is trained to professionally evaluate these aspects.

  1. 6 monthly review is not regular

We changed the term regular to “according to clinical practice”, and explained it in the text.

  1. Missing blood biochemistry

Blood biochemistry is presented in Table 1.

  1. Food diaries and recall and the inadequacies of these- if recall was over 7 days I would seriously question the validity.

We agree on the limitations of any mean to evaluate diet, which is an intrinsic limit of collecting data on nutrition.

  1. There is no statistical content

Already addressed in the original text.

Reviewer 2 Report

Comments and Suggestions for Authors

The work presented here entitled “Nutritional Counselling is effective in promoting the Mediterranean Diet and Healthy Eating in Italian Patients affected by PKU treated with Pegvaliase” by Michele Stecchi and collaboratorsis well written, clear and easy to read. The topic is cutting-edge and therefore, it adds further information to the subject area of PKU clinical research compared with other published work.

In particular, the authors using Mediterranean Diet Score and nutritional counseling to guide the transition from a medically Phe-controlled diet to a normal diet evaluated the adherence to MediDiet, when Pegvaliase enzymatic treatment is introduced.

Major

Did you dose phenylalanine (Phe) blood levels? So please add this important parameter to the table 2.  

Minor

The limitation I recognize is only the period “summer time,” and as the authors stand, “this might not allow us to capture the actual average diet during the whole year.

Author Response

The work presented here entitled “Nutritional Counselling is effective in promoting the Mediterranean Diet and Healthy Eating in Italian Patients affected by PKU treated with Pegvaliase” by Michele Stecchi and collaboratorsis well written, clear and easy to read. The topic is cutting-edge and therefore, it adds further information to the subject area of PKU clinical research compared with other published work.

In particular, the authors using Mediterranean Diet Score and nutritional counseling to guide the transition from a medically Phe-controlled diet to a normal diet evaluated the adherence to MediDiet, when Pegvaliase enzymatic treatment is introduced.

Major

Did you dose phenylalanine (Phe) blood levels? So please add this important parameter to the table 2.

We routinely evaluate Phe blood levels and were consequently added to the results.  

Minor

 The limitation I recognize is only the period “summer time,” and as the authors stand, “this might not allow us to capture the actual average diet during the whole year.

Thanks for your comment, we will plan for further studies to account for this limitation.

Reviewer 3 Report

Comments and Suggestions for Authors

I suggest a shorter title, for example:

Nutritional Counselling Promotes Mediterranean Diet Adherence in Italian PKU Patients Treated with Pegvaliase

Line 56..I suggest to include as well conc. In mg/dl: Phe levels above 600 μmol/L (10mg/dl)

Line 171-173, to reformulate, i.e.: Out of the 9 patients currently receiving Pegvaliase treatment at our center, two were excluded from the analysis criteria. One patient was in the titration phase, and the other was unable to understand or make independent decisions, with their food choices being guided by a caregiver.

Table 1: to include unit measurements for Height, Weight and to check HOMA units

For Table 3. Responses to the MDS questionnaire, please give a more detailed legend

Line 311: .. in PKU patients being…

Comments on the Quality of English Language

I suggested several correction.

Author Response

  1. I suggest a shorter title, for example: Nutritional Counselling Promotes Mediterranean Diet Adherence in Italian PKU Patients Treated with Pegvaliase

We fear that “PKU patients” might be considered potentially cause of stigmatization, as expressed by. Stangl AL, Earnshaw VA, Logie CH, van Brakel W, C Simbayi L, Barré I, Dovidio JF. The Health Stigma and Discrimination Framework: a global, crosscutting framework to inform research, intervention development, and policy on health-related stigmas. BMC Med. 2019 Feb 15;17(1):31. doi: 10.1186/s12916-019-1271-3. PMID: 30764826; PMCID: PMC6376797.

  1. Line 56..I suggest to include as well conc. In mg/dl: Phe levels above 600 μmol/L (10mg/dl)

You are right, and we included this as well.

  1. Line 171-173, to reformulate, i.e.: Out of the 9 patients currently receiving Pegvaliase treatment at our center, two were excluded from the analysis criteria. One patient was in the titration phase, and the other was unable to understand or make independent decisions, with their food choices being guided by a caregiver.

This was rephrased.

  1. Table 1: to include unit measurements for Height, Weight and to check HOMA units

Agreed, measurements were included.

  1. For Table 3. Responses to the MDS questionnaire, please give a more detailed legend

We tried to give a more detailed Legend, and also rewrote the part about scoring in Materials and Methods.

  1. Line 311: .. in PKU patients being…
  2. Treated with Pegvaliase

We fear that “PKU patients” might be considered potentially cause of stigmatization, as expressed by. Stangl AL, Earnshaw VA, Logie CH, van Brakel W, C Simbayi L, Barré I, Dovidio JF. The Health Stigma and Discrimination Framework: a global, crosscutting framework to inform research, intervention development, and policy on health-related stigmas. BMC Med. 2019 Feb 15;17(1):31. doi: 10.1186/s12916-019-1271-3. PMID: 30764826; PMCID: PMC6376797.